# Blood pressure monitoring in high-risk pregnancy to improve the detection and monitoring of hypertension (the BUMP 1 and 2 trials): protocol for two linked randomised controlled trials

Greig Dougall ![ORCID],[1] Marloes Franssen,[1] Katherine Louise Tucker ![ORCID],[1] Ly-Mee Yu,[1] Lisa Hinton ![ORCID],[1] Oliver Rivero-Arias,[2] Lucy Abel,[1] Julie Allen,[1] Rebecca Jane Band ![ORCID],[3] Alison Chisholm,[1] Carole Crawford,[1] Marcus Green,[4] Sheila Greenfield,[5] James Hodgkinson,[5] Paul Leeson,[6] Christine McCourt,[7] Lucy MacKillop ![ORCID],[8] Alecia Nickless,[1] Jane Sandall,[9] Mauro Santos,[10] Lionel Tarassenko,[10] Carmelo Velardo,[10] Hannah Wilson,[9] Lucy Yardley,[3,11] Lucy Chappell,[9] Richard J McManus ![ORCID] [1]

GD and MF are joint first authors.
LC and RJM are joint senior authors.

For numbered affiliations see end of article.

**Correspondence to**
Professor Lucy Chappell;
lucy.chappell@kcl.ac.uk

## ABSTRACT

**Introduction** Self-monitoring of blood pressure (BP) in pregnancy could improve the detection and management of pregnancy hypertension, while also empowering and engaging women in their own care. Two linked trials aim to evaluate whether BP self-monitoring in pregnancy improves the detection of raised BP during higher risk pregnancies (BUMP 1) and whether self-monitoring reduces systolic BP during hypertensive pregnancy (BUMP 2).

**Methods and analyses** Both are multicentre, non-masked, parallel group, randomised controlled trials. Participants will be randomised to self-monitoring with telemonitoring or usual care. BUMP 1 will recruit a minimum of 2262 pregnant women at higher risk of pregnancy hypertension and BUMP 2 will recruit a minimum of 512 pregnant women with either gestational or chronic hypertension. The BUMP 1 primary outcome is the time to the first recording of raised BP by a healthcare professional. The BUMP 2 primary outcome is mean systolic BP between baseline and delivery recorded by healthcare professionals. Other outcomes will include maternal and perinatal outcomes, quality of life and adverse events. An economic evaluation of BP self-monitoring in addition to usual care compared with usual care alone will be assessed across both study populations within trial and with modelling to estimate long-term cost-effectiveness. A linked process evaluation will combine quantitative and qualitative data to examine how BP self-monitoring in pregnancy is implemented and accepted in both daily life and routine clinical practice.

**Ethics and dissemination** The trials have been approved by a Research Ethics Committee (17/WM/0241) and relevant research authorities. They will be published in peer-reviewed journals and presented at national and international conferences. If shown to be effective, BP self-monitoring would be applicable to a large population of pregnant women.

## Strengths and limitations of this study

► Based on current literature, these will be the largest randomised controlled trials of blood pressure self-monitoring in pregnancy completed to date.
► The pragmatic trial designs, with broad inclusion criteria, will make findings of both linked studies applicable to routine antenatal care.
► Data on the correct targets for self-monitored blood pressure are currently sparse.
► The trials are powered for primary and some secondary outcomes but the effect of self-monitoring on rarer secondary maternal morbidity and mortality outcomes will still be uncertain.
► Women with gestational hypertension may develop this late in pregnancy making it harder to show benefit for interventions at this time.

**Trial registration number** NCT03334149

## INTRODUCTION

Raised blood pressure (BP) affects approximately 10% of pregnancies worldwide, almost half of these women develop pre-eclampsia.[1] Globally, around 15% of maternal mortality is due to pre-eclampsia so early detection and prevention are paramount.[2] In the UK, inadequate management of raised BP, in particular systolic hypertension, has previously been reported as a significant contributing factor to maternal deaths and although this has improved over the past decade, pre-eclampsia remains important.[3,4]

BP self-monitoring, where an individual measures their own BP outside of the clinical setting, allows for multiple measurements providing a better estimate of the underlying BP than intermittent clinic measurements. Such self-monitoring in pregnancy could improve both the detection and subsequent management of gestational hypertensive disorders including pre-eclampsia, while also empowering and engaging women in their own care.[5 6]

Women who are at higher risk for raised BP in pregnancy (eg, due to age or previous medical history) may require more frequent monitoring.[7 8] BP can rise rapidly in pregnancy and hypertension may go undetected in between antenatal visits.[2] Once raised BP is detected during the pregnancy, the clinical focus is on treating the hypertension, monitoring for development of pre-eclampsia and ensuring appropriate fetal surveillance.[7] Substantial resources are currently expended in monitoring such women, from the perspective of both the woman and the National Health Service (NHS).[9] If shown to be successful, self-monitoring could provide more accurate data for clinicians to use for treatment and management strategies, safely reduce the burden of multiple clinic visits for women, free up time for midwives and therefore be a cost-effective option. Self-monitoring is easy to accomplish and is now commonplace in adults with hypertension outside of pregnancy.[10 11]

BP self-monitoring is, therefore, a potentially appealing option for both the detection and management of hypertension. We have previously shown feasibility and acceptability of self-monitoring of BP in pregnancy.[5 6] This trial aims to investigate self-monitoring in two groups by undertaking two linked, parallel group, randomised trials to evaluate whether self-monitoring of BP improves the detection of raised BP during higher risk pregnancies (BUMP 1) and whether self-monitoring improves BP control in women with hypertensive pregnancy (BUMP 2).

## METHODS AND ANALYSIS
### Study design and setting
The BUMP trials are two linked, multicentre, non-masked, parallel group, randomised controlled trials investigating self-monitoring of BP during pregnancy in the UK in secondary care maternity units (sites listed on https://clinicaltrials.gov/ct2/show/NCT03334149). The work is part of a larger programme of work investigating the use of self-monitoring and testing during pregnancy.

The BUMP 1 trial is a prospective non-masked randomised controlled trial of self-monitoring of BP in pregnancy for the detection of raised BP. Women at higher risk of pre-eclampsia will be recruited through antenatal clinics. The consent process for BUMP 1 will include discussion of the transition to BUMP 2 should a woman develop raised BP.

The BUMP 2 trial is a prospective non-masked randomised controlled trial of self-monitoring of BP for the management of hypertension in pregnancy in women with chronic hypertension (CH) or gestational hypertension (GH). Women may enter this study from BUMP 1 (maintaining original randomisation) or be recruited without prior involvement. See figure 1 for an overview of the study design.

There will be an initial external pilot phase including up to 50 women in order to test trial procedures prior to the commencement of full recruitment.

### Intervention and controls groups
#### Usual care
Usual care will consist of pregnant women having their BP measured by their usual antenatal care team, and initiation and/or adjustment of medication (where

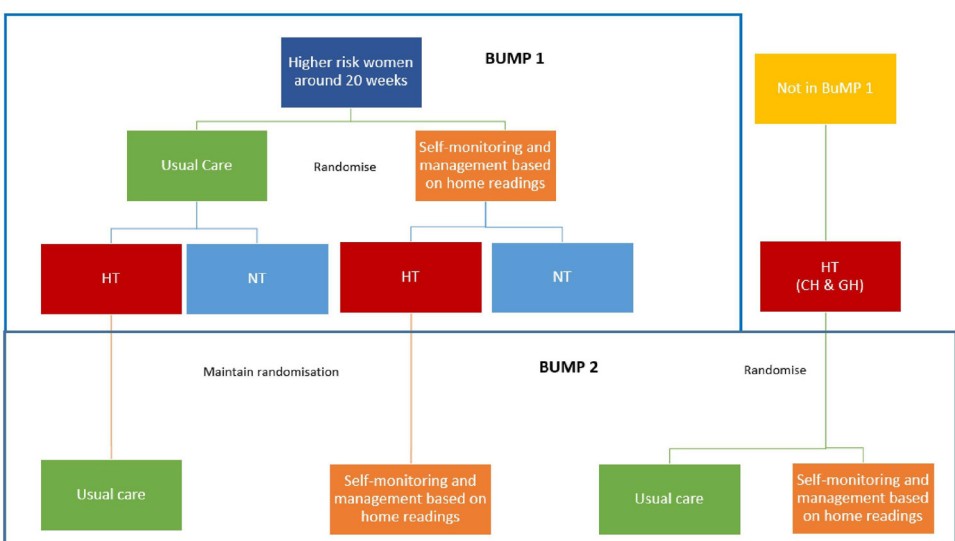

**Figure 1** Study design. BUMP 1, BP during higher risk pregnancy; CH, chronic hypertension; GH, gestational hypertension; HT, hypertension; NT, normotension.

appropriate) based on these measurements at the discretion of the healthcare professional.

## Self-monitoring

Women randomised to self-monitoring will be provided with a monitor validated for use in pregnancy and pre-eclampsia (Microlife WatchBP Home) and instructions for its use. Participants will be enrolled on the telemonitoring system, a text/app-based system and provided with clear instructions on its use. Participants can switch between the text and app system to decrease problems with connectivity for example, poor internet connection or phone signal. Access to the telemonitoring system was designed to have secure logins for the participants, their clinicians and the research team. This system was developed from our pilot work with appropriate theoretical underpinnings and designed by the University of Oxford,

Department of Engineering Science, and appropriate functionality testing was done by the development team to test different combinations of normal and abnormal BP readings and user behaviours (eg, poor adherence or numerous, unrequested readings) over a prolonged period followed by user testing with pregnant women.[6 12]

BUMP 1 participants will be asked to monitor their BP three times a week throughout their pregnancy. They will be instructed to sit quietly and comfortably for 1 min, take two readings 1 min apart and submit their second reading to the telemonitoring system. Participants will receive a guideline with colour-coded instructions (figure 2). If the second reading is outside the expected range, this will automatically trigger a request for a third reading (taken after 5 min) and persistent high or low readings will automatically trigger a message to ask the participant

| LEVEL | BLOOD PRESSURE /mmHg | ACTION |
|---|---|---|
| HIGH | SYS 150 or more OR DIA 100 or more | Your blood pressure is high<br><br>Sit quietly for 5 minutes then measure it again and send in the reading.<br><br>Contact your maternity unit for urgent assessment today (within 4 hours) and continue to monitor your BP daily. |
| RAISED | SYS 140-149 OR DIA 90-99 | Your blood pressure is raised<br><br>Sit quietly for 5 minutes then measure it again and send in the reading.<br><br>If your repeated reading is raised please contact your maternity unit within 24 hours and continue to monitor your BP daily. |
| HIGH NORMAL | SYS 135-139 OR DIA 85-89 | Your blood pressure is normal but moving towards the raised threshold<br><br>Sit quietly for 5 minutes then measure it again and send in the reading.<br><br>If your repeat reading is still high-normal, please monitor your blood pressure daily. |
| NORMAL | SYS 110-134 OR DIA 70-84 | Your blood pressure is normal.<br><br>Continue blood pressure monitoring and your current care |
| LOW | SYS 109 or less AND DIA 69 or less | Your blood pressure is low. Repeat once more in 5 minutes.<br><br>If you are taking blood pressure medication, contact your maternity unit within 24 hours or within 4 hours if you feel unwell (e.g. dizzy or faint).<br><br>If you are not taking medication and you are feeling well this blood pressure does not need any further action. |

**Figure 2** BP interpretation chart for BUMP 1. BUMP1, BP during higher risk pregnancy.

| LEVEL | BLOOD PRESSURE /mmHg | ACTION |
|---|---|---|
| HIGH | SYS 150 or more OR DIA 100 or more | Your blood pressure is high<br><br>Sit quietly for 5 minutes then measure it again and send in the reading.<br><br>Contact your maternity unit for urgent assessment today (within 4 hours) and continue to monitor your BP daily. |
| RAISED | SYS 140-149 OR DIA 90-99 | Your blood pressure is raised<br><br>Sit quietly for 5 minutes then measure it again and send in the reading.<br><br>If your repeated reading is raised please contact your maternity unit within 24 hours and continue to monitor your BP daily. |
| NORMAL | SYS 110-139 OR DIA 70-89 | Your blood pressure is normal.<br><br>Continue blood pressure monitoring and your current care |
| LOW | SYS 109 or less AND DIA 69 or less | Your blood pressure is low. Repeat once more in 5 minutes.<br><br>If you are taking blood pressure medication, contact your maternity unit within 24 hours or within 4 hours if you feel unwell (e.g. dizzy or faint).<br><br>If you are not taking medication and you are feeling well this blood pressure does not need any further action. |

**Figure 3**  BP interpretation chart for BUMP 2. BUMP 2, BP during hypertensive pregnancy.

to contact their local maternity unit. If women have persistently high readings, they will be asked (via the telemonitoring system) to undertake daily readings for the rest of their pregnancy.

BUMP 2 participants will be asked to monitor their BP daily throughout their pregnancy. They will be instructed to take two measures (as above) and asked to submit their second reading to the telemonitoring system. Participants will receive a guideline with colour coded instructions (figure 3). High or low second readings will automatically trigger a request for a third reading (taken after 5 min) and persistent high or low readings will automatically trigger a message for the participant to contact their local maternity unit.

On the basis of the results of our pilot work and recent systematic review, the same thresholds will be used for self-monitoring as clinic BP to trigger action (ie, 140/90 mm Hg), therefore, resulting in alerts to participating women via the telemonitoring system.[6 13]

To improve adherence to the intervention participants will receive weekly motivational messages, and these messages will be sent to women in the intervention group via the telemonitoring system (via the app or text message) based on the woman's preference. These are designed to provide support and education throughout the trial and were developed during our pilot work with

patient and public involvement. Messages will be selected at random from a pool of messages (approved by the ethics committee). To monitor adherence and for safety the local clinical team will also receive warnings in real time via the telemonitoring system if a participant misses submitting readings, allowing them to approach the participant to troubleshoot any potential issues.

Participants will be provided with instructions in the use of the telemonitoring system at the baseline visit. Participants will manually send their readings from the BP monitor to a centralised database, secured behind NHS firewalls, using a free SMS text message or app with web-based data entry backup. Reminders and triggered messages will be received by participants according to a rule-based algorithm developed with the clinical team. Participants in BUMP 1 will receive a reminder after 4 days without submitting a reading and then one more reminder the following day if no further readings are entered. Participants in BUMP 2 will receive a reminder after two missed days of readings, with a further reminder the following day if no further readings are entered.

Each participant has the right to discontinue the intervention or withdraw from the trial at any time. In addition, an investigator may withdraw a participant from the trial at any time if the investigator considers it necessary for any reason including: ineligibility (either arising

during the trial or retrospectively having been overlooked at screening), an adverse event (AE), which results in inability to continue to comply with trial procedures, withdrawal of consent and lost to follow-up. If a woman discontinues the intervention, withdraws or is withdrawn from the study at any point, her usual antenatal care will continue as all study procedures are additional rather than in place of usual care.

Women who wish to discontinue the BP self-monitoring intervention will be asked if they are willing to participate in study follow-up. All data collected to the point of withdrawal will be retained in the study database. Unless a participant specifically withdraws consent, notes review will be conducted if possible, even where an individual has been lost to follow-up.

## Study outcomes
### Outcome definitions
- The onset of hypertension/high BP will be defined as the earliest date recorded of sustained systolic BP ≥140 and/or diastolic BP ≥90 mm Hg from any community or hospital setting recorded in the notes review.
- Sustained is defined as at least two readings within 1 week (168 hours), with no minimum time between readings needing to be recorded.
- The 'earliest date recorded' will be for the second of these two readings.
- Additionally, a new prescription of antihypertensive medication for raised BP will be taken as a diagnosis of hypertension/high BP (in order to capture diagnoses where medication has been started but two raised BPs have not been recorded in the notes review for whatever reason). In this case, the first date of prescription will be listed as the date of diagnosis.

### Primary outcome for BUMP 1
The primary outcome for BUMP 1 is the difference in time to a recording of raised BP by a healthcare professional between the usual care and self-monitoring groups.

### Secondary outcomes for BUMP 1
Secondary outcomes for BUMP 1 include analysis of the difference between usual care and self-monitoring groups as per box 1:

### Primary outcome for BUMP 2
The primary outcome for BUMP 2 is the difference in mean systolic BPs, recorded by healthcare professionals from baseline to delivery, between usual care and self-monitoring groups.

### Secondary Outcomes for BUMP 2
Secondary outcomes for BUMP 2 include analysis of the difference between usual care and self-monitoring groups as per box 1.

### Participant timeline
Members of the research team will provide a full verbal explanation and written description of the trial to women

---

**Box 1  Secondary outcomes for blood pressure (BP) during higher risk pregnancy (BUMP 1) and BUMP 2**

**Maternal outcomes**
- Severe hypertension (systolic BP ≥160 mm Hg and/or diastolic BP ≥110 mm Hg).
- Pre-eclampsia or gestational hypertension.
- Serious maternal complications including eclampsia, transient ischaemic attack or stroke, pulmonary oedema, renal failure, blood transfusion, HELLP syndrome, liver involvement, haematological involvement and death.
- Onset of labour.
- Change in maternal quality of life from randomisation (measured using the EuroQoL 5 Dimensions 5 Levels (EQ-5D-5L) questionnaire).[16]
- Diastolic BP.*
- Area under the BP curve.*
- Proportion of systolic readings above 140 mm Hg.*

**Perinatal Outcomes**
- Stillbirths.
- Early neonatal deaths.
- Gestation at delivery.
- Mode of delivery.
- Birth weight of the baby, including centile.
- Small for gestational age infants (<10th and <3rd centile).
- Neonatal unit admissions, including length of stay.

**Process outcomes**
- Change in health behaviours (questionnaire).
- Fidelity to monitoring schedule.
- Change in State Trait Anxiety Inventory short form 6 anxiety questionnaire.[34]
- Health service costs.
- Cost per maternal quality-adjusted life year gained over trial period.
- Change in adherence to medication (MARS questionnaire).*[35]
- Qualitative data gathered from participating women and healthcare professionals.

Outcomes common to both trials apart from. *Only assessed on BUMP 2 participants.

---

who meet the inclusion criteria (box 2 and 3). The woman will be given sufficient time to consider the information, and to decide whether she will participate in the trial. Written informed consent will be sought from the woman and taken by an appropriately trained healthcare professional. Eligibility will be assessed and baseline data, including all demographics, maternal demographics and patient questionnaires (completed by the woman), will be entered on a web-based database by members of the research team. Participants will be randomised and instructed in the use of self-monitoring and provided with a BP monitor or instructed to follow usual care. Women randomised to usual care will not be prevented from self-monitoring although we will ask all women at final follow-up if they have self-monitored during the trial. Participants on both arms will follow their routine care clinic visits until delivery. Participants will be contacted at around 30 weeks gestation (2 weeks after baseline if participants were recruited after 30 weeks) and asked either in person/over the phone/by post/or via email to

> **Box 2  Blood pressure during higher risk pregnancy (BUMP 1) inclusion and exclusion criteria**
>
> **Inclusion criteria**
> ► Participant is willing and able to give informed consent for participation in the trial.
> ► Pregnant woman, aged 18 years or above between $16+^0$ and $24+^0$ weeks.
> ► Able and willing to comply with trial requirements.
> ► Willing to allow her general practitioner and consultant, if appropriate, to be notified of participation in the trial.
> ► At higher risk for hypertension in pregnancy/pre-eclampsia defined as one or more of the following risk factors:
> – Age 40 years or older.
> – Nulliparity.
> – Pregnancy interval of more than 10 years.
> – Family history of pre-eclampsia.
> – Previous history of pre-eclampsia or gestational hypertension.
> – Body mass index 30 kg/m$^2$ or above at booking.
> – Chronic kidney disease.
> – Twin pregnancy.
> – Prepregnancy diabetes (type 1 or 2)
> – Autoimmune disease (eg, systemic lupus erythematosus or antiphospholipid disease).
>
> **Exclusion criteria**
> ► Chronic hypertension.

complete patient questionnaires. The remainder of antenatal care, in particular the timing and mode of delivery, will be left to the discretion of the responsible clinician. A medical notes review will be undertaken following the primary discharge for mother and baby. If a woman and/or her baby is still in hospital at 2 months from delivery or 2 months from her estimated date of delivery (whichever

> **Box 3  Blood pressure (BP) during hypertensive pregnancy inclusion and exclusion criteria**
>
> **Inclusion criteria**
> ► Women with chronic hypertension (defined as sustained systolic BP ≥140 mm Hg and/or diastolic BP ≥90 mm Hg, present at booking or before 20 weeks gestation, or receiving antihypertensive treatment outside pregnancy and/or at time of referral), recruited up to $37+^0$ weeks gestation.
>
> **OR**
> ► Women with gestational hypertension after 20 weeks gestation (defined as sustained systolic BP ≥140 mm Hg and/or diastolic BP ≥90 mm Hg), recruited at $20+^0$ to $37+^0$ weeks gestation.
>
> **AND**
> ► Participant is willing and able to give informed consent for participation in the trial.
> ► Woman aged 18 years or above.
> ► Willing to allow her general practitioner and consultant, if appropriate, to be notified of participation.
>
> **Exclusion criteria**
> ► Anticipated inpatient admission considered likely to lead to imminent delivery (within the next 48 hours).

is longer), the admission time will be censored and data collected up to that point. Self-monitoring of BP will finish at admission in established labour or the end of pregnancy, whichever is sooner and participants will continue with standard postnatal care. Participants will be contacted at around 8 weeks postdelivery and asked either in person/over the phone/by post/or via email to complete patient questionnaires and arrange for collection of the loaned BP monitor if not returned at delivery. A schedule of participant enrolment, interventions and assessments in the trial is shown in table 1 and a flowchart of study visits is shown in figures 4 and 5.

### Sample size considerations
#### BP during higher risk pregnancies

The sample size was determined using a two-stage simulation process, which modelled how many women would be expected to develop hypertension and how long time to detection would take in these women, using data from our pilot work in the BUMP study. Assuming 16% of women develop hypertension, and an SD of 40 days to detection of raised BP in both groups, a sample size of 2262 (1131 per group) will allow detection of an effect size of 12 day's difference in time to detection of raised BP in pregnancy between self-monitoring and control groups (the primary outcome of BUMP 1), with 90% power and 5% level of significance (two sided) and assuming a 15% attrition rate. If the SD is 45 days, then this sample size will allow detection of a difference of 14 days with more than 90% power and if the SD is 50 days then it will be sufficient to detect a difference of 16 days in time to detection of raised BP in pregnancy also with 90% power. Of the 2262 women recruited to BUMP 1, around 362 women are expected to develop hypertension. We will recruit a minimum of 2262 women to ensure adequate power. The simulation was carried out using R V.3.1.2. (https://www.r-project.org/).

#### BP during hypertensive pregnancy

A sample size of 256 per group will be sufficient to detect a 5 mm Hg difference in systolic BP between groups, accounting for 15% attrition and an SD of 16 mm Hg, based on data from the BUMP pilot and PELICAN studies, inflated from 14 mm Hg because BUMP 2 will include hypertensive women from BUMP 1 and those with chronic or GH not previously randomised in BUMP 1.[6 14] The sample size was calculated using NCSS PASS V.12.0.

Women randomised in BUMP 1 who develop hypertension will remain in their randomisation groups and move seamlessly into the BUMP 2 trial. This will occur automatically via the telemonitoring system and their inclusion in BUMP 2 assessed retrospectively. The precise numbers and proportion of women migrating from BUMP 1 to BUMP 2 will not be fixed. We will aim to recruit a minimum of 512 women directly into the BUMP 2 trial in addition to those that migrate in order to ensure adequate power: women who have CH or have developed

**Table 1** Schedule of interventions and assessments

| Study visit | Study period (16 weeks to 8 weeks postdelivery) | | | |
| | Trial entry/baseline | Antenatal follow-up | Notes review | Postnatal follow-up |
| Time point | 20±4 weeks gestation | 30±3 weeks gestation | 40+4 weeks (postdelivery) | 40+8 weeks (postdelivery) |
| --- | --- | --- | --- | --- |
| Informed consent | X | | | |
| Eligibility assessment | X | | | |
| Demographics; including age, race, education | X | | | |
| Maternal demographics; including gestation, EDD and parity | X | | | |
| Patient Questionnaires; including EQ-5D-5L and STAI | X | X | | X |
| Randomisation | X | | | |
| Recording home BP | Self-Monitoring group will record and submit reading between baseline and delivery | | | |
| Recording clinic BPs | | | X | |
| Maternal and perinatal outcomes | | | X | |
| Secondary outcome data | | | X | |
| Return of BP monitor | | | BP monitors will be returned at/or postdelivery | |

BP, blood pressure; EDD, Estimated Delivery Date; EQ-5D-5L, EuroQoL 5 Dimensions 5 Levels; STAI, State-Trait Anxiety Inventory.

GH and are not taking part in the BUMP 1 trial will be randomised to either usual care using clinic BP to guide treatment or BP self-monitoring with telemonitoring and automated feedback.

Both trials will be undertaken in approximately 15 maternity units in England to achieve the sample size in the anticipated time frame. Our previous pilot trial has confirmed that women and clinicians are willing to participate in this type of randomised controlled trial and we have used recruitment estimates from this trial, together with trial management experience and expertise of the coinvestigator group to inform strategies to meet the required target size.[6]

## Data management
An online clinical data management system will be used, with all clinical data entered directly onto the electronic case report forms (CRFs) and stored on a study-specific database. Paper backup forms will be available if the online database cannot be accessed. Data collected on paper forms will be entered on to the database at the earliest opportunity, by single data entry. Patient-completed questionnaires will be completed directly on to the system by the participant wherever possible, with manual single data entry of paper copies if required. Telemonitored mHealth data are stored on a centralised database, secured behind NHS firewalls, designed and maintained by the University of Oxford, Department of Engineering Science.

## Randomisation
Women who agree to participate in the trial (BUMP 1 or BUMP 2) will be randomised on a 1:1 ratio either to BP self-monitoring or usual care by the recruiting researcher. An independent statistician will generate a randomisation sequence list for each trial, using permutated varying blocks and stratified by recruitment site and parity. The generated schedules were then imported to the randomisation module within the online data management system for site to carry out the randomisation. Women will be randomised by the recruiting researcher using the online data management system. Women who develop hypertension during BUMP 1 and migrate to BUMP 2 will stay in their original randomisation group. A manual telephone based backup randomisation system will be used in the event the online system is not available.

## Statistical analysis
### General consideration for both trials
The primary statistical analysis will be carried out on the basis of intention to treat (ITT). Thus, after randomisation participants will be analysed according to their allocated treatment group irrespective of what treatment they actually receive. We will endeavour to obtain full follow-up data on every participant to allow full ITT analysis, but we will inevitably experience the problem of missing data due to withdrawal, lost to follow-up, or non-response questionnaire items. The results from the trial will be prepared as comparative summary statistics

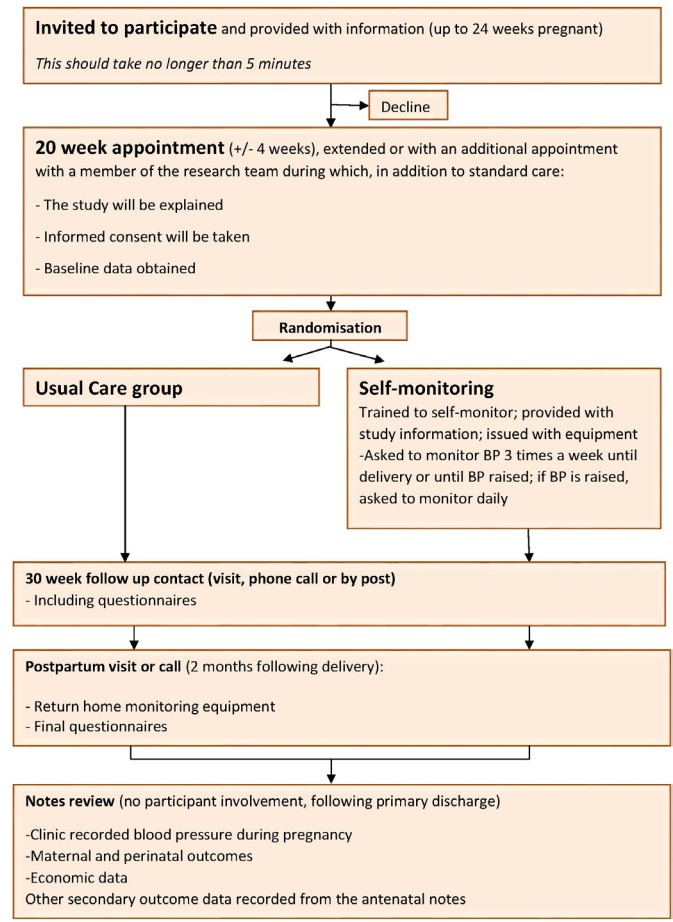

**Figure 4** BUMP 1 study visit chart. BP, blood pressure; BUMP 1, BP during higher risk pregnancy.

(eg, difference in means) with 95% CIs. The study results will be reported in accordance with the Consolidated Standards of Reporting Trials 2010 statements and full detailed statistical analysis plans will be prepared before the final analysis by an independent statistician.

## BP during higher risk pregnancy

The primary analysis will determine whether there is a difference in the time to diagnosis between the randomised groups (usual care vs self-monitoring) and will be performed using a two-part model, such as the hurdle model.[15] The model takes into account participants who developed a raised BP or not and the time to raise BP, adjusting for stratification factors. This method has the advantage that all women who are recruited to BUMP 1 will contribute to the primary analysis. Continuous secondary outcomes, such as birth weight and length of stay, will be analysed by means of regression method adjusting for stratification factors. Binary secondary outcomes, such as the development of severe hypertension, development of any complications and stillbirth rate, will be analysed by means of a log binomial model.

## BP during hypertensive pregnancy

An ITT analysis will include women recruited to BUMP 1 who become hypertensive and women recruited de novo

to BUMP 2 with GH and CH analysed separately. The primary analysis for BUMP 2 will in each case compare BPs between the intervention and control groups. Analysis of the BP outcome will be by means of a linear mixed-effects model, which can accommodate data where participants have repeated measurements, and also accounts for missing data (assuming data are missing at random). Each of a participant's BP measures will be modelled by means of a linear mixed-effects model, including random effects for participants and assuming an unstructured variance covariance matrix between measurements from the same participant. Time of BP measurement and randomised group, as well as their interaction term, will be included as fixed effects in the model. A fixed effect will also be included for whether the participant was from BUMP 1 or not in the GH analysis. As a sensitivity analysis inclusion from BUMP 1 or not will be tested as a moderator of the treatment effect. By means of appropriate contrasts, the difference in mean BP of participants in the randomised groups from both BUMP 1 and de novo recruitment will be compared.

BP load and area under the curve will be analysed as continuous outcomes by means of a linear regression model, adjusting for whether the participant originated from BUMP 1. Continuous secondary outcomes, such as

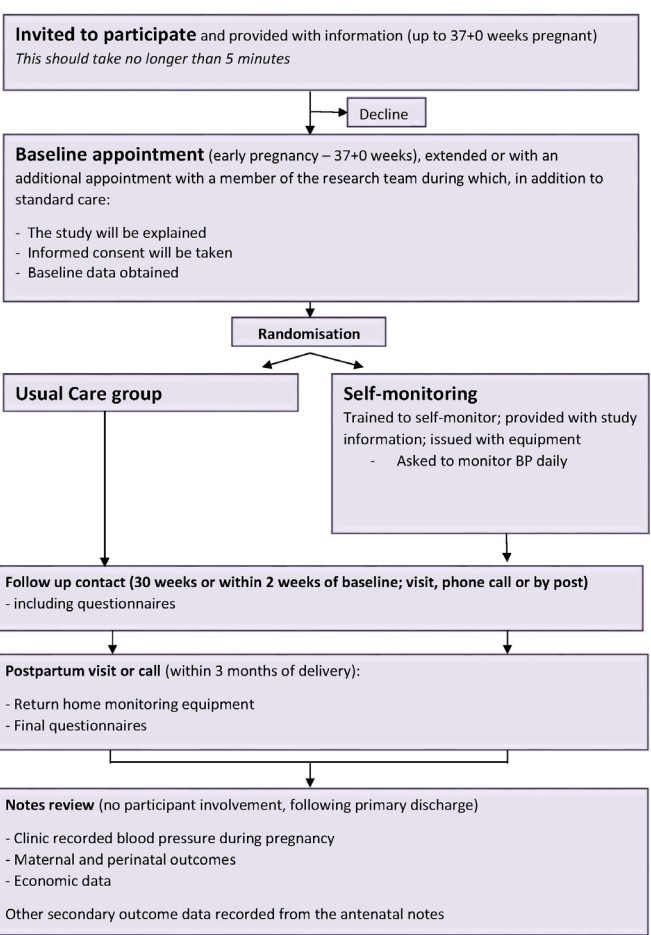

**Figure 5** BUMP 2 study visit chart. BP, blood pressure; BUMP 2, BP during hypertensive pregnancy.

birth weight and length of stay, will be analysed by means of regression model. Binary secondary outcomes, such as the development of severe hypertension, incidence of complications and incidence of stillbirth, will be analysed by means of a log binominal model.

## Subgroup analyses

For both BUMP 1 and BUMP 2 prespecified subgroups include eligibility for aspirin prescription to prevent pre-eclampsia (as per National Institute for Health and Care Excellence); gestational age at baseline (≤20 weeks vs >20 weeks), parity (0 vs ≥1). In addition, a sensitivity analysis will compare those women with GH that are recruited de novo into BUMP 2 with those that migrate from BUMP 1.

## Economic evaluation

The economic evaluation component aims to determine whether BP self-monitoring in addition to usual care represents value for money compared with usual care alone in a population of higher risk pregnant women (BUMP 1) and women with a diagnosis of hypertension during pregnancy (BUMP 2). Two separate within-trial cost-effectiveness analyses will be carried out alongside BUMP 1 and BUMP 2. In addition, a decision analytical model will be built to determine the long-term cost-effectiveness of self-monitoring in pregnancy. A UK NHS perspective will be adopted in all analyses.

The within-trial cost–utility analyses will use individual patient-level data with a time horizon up to 2 months following delivery. The main health outcome measure in the economic evaluation will be maternal quality-adjusted life years (QALYs) over the trial period. The calculation of a QALY profile for each woman will be informed using data from the health-related quality of life instrument EQ-5D-5L that will be collected at baseline, 30 weeks gestation and 8 weeks after birth.[16] A linear change between utility measures at each time period will be assumed when deriving an individual's QALY profile. Relevant healthcare resource use during the pregnancy for mothers and their fetuses will include antenatal, intrapartum and postnatal care. Antenatal care utilisation will include primary and secondary care visits including hospital admissions. Intrapartum and related postnatal care before primary hospital discharge will include major procedures undertaken, length of stay and transfers for mothers and their babies. Primary and secondary care resource utilisation during the trial will be collected using case note review including routine and additional clinic visits to midwives and general practitioners (GPs). Unit costs will be extracted from primarily national sources.[17–19] The cost of using an automated monitor along with the telemonitoring system will be estimated and included in the cost analysis of the self-monitoring group in each trial. Mean maternal QALYs and costs (and associated uncertainty) will be estimated parametrically in the intervention and usual care group in each of the trials. If necessary, we will use multiple imputation methods to deal with missing EQ-5D-5L and resource use data over the trial period and

will follow current guidance to conduct and report such analysis.[20]

The long-term decision analytical model will include a decision tree to represent women's pathways up to the first 8 weeks postdelivery and a Markov model representing the history of disease of women with raised BP after that point, using annual cycles. The structure of the model will be based on models identified via a systematic review of previous economic evaluations of hypertensive patients in pregnancy (currently in progress with PROSPERO number: 123881). Observed risk factors, quality of life and healthcare resource utilisation for randomised participants in BUMP 1 and BUMP 2 will be used to inform the characteristics of a hypothetical cohort entering the model for each treatment pathway. Transition probabilities indicating movement across health states will be informed by cardiovascular risk predictions modified using the literature on risk following hypertensive pregnancy. Costs incurred annually in each health state will be obtained from the literature differentiating between acute and chronic disease health states. Health-related quality of life associated with each health state in the model will also be obtained from the literature.

Both the within-trial cost-effectiveness analysis and long-term model will be reported using current reporting standards for economic evaluations.[21] Mean differences in costs and QALYs between intervention and usual care arms in each trial and the model will be combined to generate an incremental cost-effectiveness ratio (ICER), presented as cost per QALY gained. Uncertainty around ICERs will be handled using CIs (if appropriate) and cost-effectiveness acceptability curves. Uncertainty in the long-term model will be assessed using probabilistic sensitivity analysis. Recent guidance will also be used to report and interpret the results from the decision analytical model.[22]

## Process evaluation (qualitative and quantitative)

Alongside the trial, we will undertake a process evaluation to assess both quantitative (including the outcomes listed in box 1 above) and qualitative aspects. Intervention fidelity will be assessed for women and participating professionals. This will include for the women: examining persistence with self-monitoring and comparing submitted readings and monitor downloads for a subset of women. For the professionals, a staff survey will assess practice in terms of utilisation of self-monitored BP readings during the trial.

In addition to the quantitative process evaluation, a qualitative study will be conducted within the trial to understand how BP self-monitoring in pregnancy is implemented and its acceptability in daily life and routine clinical practice. In-depth interviews will be completed with a sample of participating women in both BUMP 1 and BUMP 2 and healthcare professionals.[23]

Participating women: The study aims to include approximately 40 pregnant women across five sites, including a sample of up to eight women who decide not to take part or discontinue from the intervention and are willing to

be interviewed.[24] The interviews with women will look to understand real-life implementation issues with BP self-monitoring. Women who have consented to contact at main trial enrolment will be contacted by phone/email/in person regarding the interviews and provided with a patient information sheet (PIS); they will be consented separately prior to the start of the interview. Additionally, an ethnographic study will follow a sample of 10–15 women who monitor their BP.[25] This ethnographic study will include longitudinal interviewing and non-participatory observations to understand in detail how women engage with home monitoring over time and whether and how home BP readings are incorporated into clinical encounters.[26 27]

Healthcare professionals: Interviews will be conducted with 30–40 midwives and obstetricians who are involved in BUMP 1 and BUMP 2.[24] The aim is to understand how self-monitoring is operationalised in practice and provide key information for the process evaluation.

Detailed field notes will be taken of observations. Interviews will be audio recorded so that content can be transcribed. Data collection and analysis will be guided by theory, including social cognitive theory, a 'technology in practice' perspective and normalisation process theory as interpretative frameworks to understand how the trial intervention 'fits' within existing practices and systems, and how it becomes embedded (or not) in women's lives and healthcare professionals' routine work.[28–31] Inductive and deductive approaches to categorising and coding the data will be combined, drawing on a priori theoretical concepts (including, g, self-efficacy, unexpected use of technology and cognitive participation) while remaining sensitive to themes that emerge from the data themselves. An iterative analytical process will be employed to map the range of phenomena and identify associations between themes with the aim of illuminating the mechanisms that underpin the outcomes of the intervention.[32] QSR NVivo will be used to support the organisation and retrieval of the qualitative data.[33]

## Data monitoring

Regular monitoring will be performed by the trial team according to Good Clinical Practice (GCP) and Clinical Trial Unit standard operating procedures. Data will be evaluated for compliance with the protocol and accuracy in relation to source documents. Following written standard operating procedures, the monitors will verify that the clinical trial is conducted and data are generated, documented and reported in compliance with the protocol, GCP and the applicable regulatory requirements.

A trial steering committee (TSC) and data monitoring committee (DMC) will be convened and review trial progress every 6 months. The TSC will provide overall supervision of the trial and ensure its conduct is in accordance with the principles of GCP and the relevant regulations. The TSC will agree the trial protocol and provide advice to the investigators on all aspects of the trial. The TSC will include members who are independent of the investigators, in particular an independent chairperson. The independent DMC will inform the TSC regarding the accruing trial and safety data, to ensure trial site staff and participants are aware of any relevant safety information and to advise the TSC regarding the appropriateness of continuation of the trial. Any consideration of the need for termination of the trial will be on the advice of the DMC and TSC who will consider the need for prospective criteria for termination.

## Harms/potential risks

During these randomised controlled trials, participants will continue to receive 'usual care' regardless of randomisation group, and thus we anticipate that the potential risks are low. Particular issues include the possibility of increased anxiety due to the study. Training of participants will cover repeated measurements in the case of unusually high or low readings. The participant guideline/booklet will give clear advice to women to contact the antenatal care team or other healthcare professional (eg, GP) in the case of persistent high or low readings. The telemonitoring system will automatically provide this advice when high or low readings are sent in. Women will continue to be seen by their clinical teams (midwives/GPs/obstetricians) at frequencies chosen by their clinicians throughout, regardless of randomisation group.

Participants will be advised to seek immediate medical help if they experience any symptoms of pre-eclampsia, regardless of their BP readings.

It is not anticipated that the study intervention (self-monitoring of BP) should result in any AEs but collection of such events will be included in case such events occur so that they can be considered for causal links to the study. Only AEs that are clinically judged (by the supervising site principal investigator) as being caused by the trial intervention will be reported to the Clinical Trials Unit to raise appropriately to the sponsor/Research Ethics Committee that gave favourable opinion. In addition, any maternal death or stroke, fetal loss or neonatal death will be reported to the DMC regardless as to whether they are judged related. Side effects as stated in the British National Formulary will not be reported as AEs.

## ETHICS AND DISSEMINATION
### Ethics approval and consent to participate

Any subsequent protocol amendments will be agreed with both sponsor and ethics committee prior to implementation. The study sponsor reviewed and ensured all indemnity and insurance requirements for the trial were in place prior to the start of recruitment which would operate in the event of any participant suffering harm as a result of their involvement in the research. NHS indemnity operates in respect of the clinical treatment that is provided. Participants will provide written informed consent prior to enrolment. An independent trial steering group will monitor study progress assisted by an independent DMC and will periodically review the study.

## Confidentiality

The trial staff will ensure that the participants' anonymity is maintained in the trial database. The participants will be identified only by a participant identification number on all trial documents and any electronic database. Women's identifiers (names, address and phone number) will be held securely and separately from the CRFs where they are needed to contact participants on an ongoing basis, for example, for follow-up and return of BP monitor. Access to these data will be strictly on a need to know basis. All documents will be stored securely and only accessible by trial staff and authorised personnel. The study will comply with the Data Protection Act and General Data Protection Regulation 2018, which requires data to be anonymised as soon as it is practical to do so.

## Patient and public involvement

Methods of patient approach, PIS and consent form were all reviewed by the patient representatives prior to formal approval. The relevant group for women with pre-eclampsia, 'Action on Pre-Eclampsia' are formally represented through their chief executive officer (MG) who is a coinvestigator. The TSC includes an independent patient representative who is jointly responsible for overseeing the conduct of the trial with the rest of the committee.

## Dissemination

All research outputs from this work will be published in peer-reviewed journals, presented at scientific conferences and lay and social media (eg, Twitter, blogs). All trial data are the property of the chief investigator and will be stored until the end of the study, database lock and final analysis. All research data and documentation will be appropriately archived for at least 5 years. 'Patient friendly' study summary documents and infographics will be made available to all participants at the end of the trial via the study website.

## DISCUSSION

This article describes the protocol for the BUMP trials. These randomised controlled trials will assess whether self-monitoring of BP during pregnancy, over and above usual care, can improve the detection of raised BP during pregnancy and whether self-monitoring can improve the control of BP in pregnancy hypertension.

Additional work will show whether self-monitoring BP is acceptable to pregnant women and their healthcare professionals and if it is cost-effective compared with usual care.

The results of the trial will provide data on the effects of self-monitoring and telemonitoring, over and above usual care, in pregnant women at risk for pre-eclampsia and women with hypertension.

If self-monitoring of BP is found to be successful in the detection of hypertension in pregnancy and control of BP in pregnancy hypertension, then it would be applicable to a large group of pregnant women in the UK but also worldwide.

## Current trial status

The trials commenced recruitment on 11 September 2017 (external pilot) with main recruitment starting on (22 November 2017) and recruitment is planned to continue until end of September 2019. Data collection will continue to mid-2020.

**Author affiliations**
[1]Nuffield Department of Primary Care Health Sciences, University of Oxford, Oxford, UK
[2]National Perinatal Epidemiology Unit (NPEU), Nuffield Department of Population Health, University of Oxford, Oxford, UK
[3]Academic Unit of Psychology, University of Southampton, Southampton, UK
[4]Action on Pre-eclampsia, Evesham, UK
[5]Primary Care Clinical Sciences, Institute of Applied Health Research, University of Birmingham, Birmingham, UK
[6]Cardiovascular Clinical Research Facility, Division of Cardiovascular Medicine, University of Oxford, Oxford, UK
[7]Centre for Maternal & Child Health Research, School of Health Sciences, City University, London, UK
[8]NIHR Oxford Biomedical Research Centre, Oxford University Hospitals NHS Foundation Trust, Oxford, United Kingdom
[9]Department of Women and Children's Health, Kings College, London, London, UK
[10]Institute of Biomedical Engineering, Department of Engineering Science, University of Oxford, Oxford, UK
[11]School of Psychological Science, University of Bristol, Bristol, UK

**Twitter** Carmelo Velardo @2dvisio

**Acknowledgements** Lucy Mackillop was supported by the NIHR Oxford Biomedical Research Centre.

**Contributors** RJM had the original idea and applied for funding with KLT, LC, L-MY, LH, OR-A, CC, MG, SG, JH, PL, CM, LM, JS, LT and LY. MF, GD, RJM, LC and KLT wrote the first draft. AN and L-MY provided the sample size calculations and statistical analysis section. RJM, KLT, LH, RJB, LY, MS and CV worked on developing the intervention. CC and HW are coordinating research midwives. LH and AC provided the qualitative section. OR-A and LA provided the economic evaluation section. GD is the trial manager (previously MF) and JA is the senior trial manager. All authors subsequently critically edited the manuscript. RJM will be guarantor for the manuscript. All authors have read and approved the final manuscript.

**Funding** This work is funded from a National Institute for Health Research (NIHR) Programme grant for applied research (RP-PG-1209-10051) and an NIHR Professorship awarded to RJM (NIHR-RP-R2-12-015). RJM and KLT receive funding from the National Institute for Health Research (NIHR) Collaboration for Leadership in Applied Health Research and Care Oxford at Oxford Health NHS Foundation Trust. JS is a National Institute for Health Research (NIHR) Senior Investigator and supported by the National Institute for Health Research (NIHR) Collaboration for Leadership in Applied Health Research and Care South London (NIHR CLAHRC South London) at King's College Hospital NHS Foundation Trust. Service support costs will be administered through the NIHR Clinical Research Network.

**Disclaimer** The views expressed in this publication are those of the authors and not necessarily those of the NHS, the NIHR or the Department of Health and social care.

**Competing interests** RJM has previously received BP monitors from Omron for research purposes. The BP monitors for the current trials were purchased from the manufacturer (Microlife) at commercial prices.

**Patient consent for publication** Not required.

**Ethics approval** The protocol, informed consent form, participant information sheet and all other participant-facing material have been approved by the Research Ethics Committee (West Midlands - South Birmingham: ref 17/WM/0241), host institution(s) and Health Research Authority.

**Provenance and peer review** Not commissioned; externally peer reviewed.

## ORCID iDs

Greig Dougall http://orcid.org/0000-0001-9751-1998
Katherine Louise Tucker http://orcid.org/0000-0001-6544-8066
Lisa Hinton http://orcid.org/0000-0002-6082-3151
Rebecca Jane Band http://orcid.org/0000-0001-5403-1708
Lucy MacKillop http://orcid.org/0000-0002-1927-1594
Richard J McManus http://orcid.org/0000-0003-3638-028X

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
