## [Reviewer comments · BMJ Open]

ARTICLE DETAILS

TITLE (PROVISIONAL)	Blood pressure monitoring in high risk pregnancy to improve the detection and monitoring of hypertension (The BUMP 1&2 trials): protocol for two linked randomised controlled trials.
AUTHORS	Dougall, Greig; Franssen, Marloes; Tucker, Katherine Louise; Yu, Ly-Mee; Hinton, Lisa; Rivero-Arias, Oliver; Abel, Lucy; Allen, Julie; Band, Rebecca Jane; Chisholm, Alison; Crawford, Carole; Green, Marcus; Greenfield, Sheila; Hodgkinson, James; Leeson, Paul; McCourt, Christine; MacKillop, Lucy; Nickless, Alecia; Sandall, Jane; Santos, Mauro; Tarassenko, Lionel; Velardo, Carmelo; Wilson, Hannah; Yardley, Lucy; Chappell, Dr Lucy; McManus, Richard J

VERSION 1 – REVIEW

REVIEWER	Jacob Alexander Lykke Dept. of Obstetrics, Rigshospitalet, University of Copenhagen, Copenhagen, Denmark,
REVIEW RETURNED	20-Oct-2019

GENERAL COMMENTS	The authors are to be applauded for this study protocol. I am looking forward to the results are presented.
---

REVIEWER	Stefan Rahr Wagner Aarhus University, Department of Engineering, Denmark
REVIEW RETURNED	07-Nov-2019

GENERAL COMMENTS	I have reviewed the protocol for the BUMP 1&2 trials, which are two NIHR funded trials seeking to assess the effectiveness and cost effectiveness of self-monitoring of blood pressure in pregnancy using a multicenter RCT approach. A very relevant research topic to pursue. Specifically, BUMP 1 assesses whether self-monitoring of blood pressure can detect raised blood pressure earlier than routine clinic monitoring which is highly relevant, while BUMP 2 assesses whether self-monitoring of blood pressure alongside usual care leads to lower blood pressure in hypertensive pregnancy. Both of these studies are highly relevant to perform as not much work has been made in this area. Also, the economical evaluation is highly relevant, as a broader home screening of blood pressure is expensive in terms of equipment and personnel resources for follow-up. Although the protocol is quite lengthy (5147 words!), this can be explained by the combined BUMP 1 and 2 protocols, both of which are very ambitious with both quantitative and qualitative studies, appearing in one protocol. While this is arguably reasonable, due to the two studies being tightly connected, it also reduces the clarity of the overall protocol. There are many factors to consider.
---

	Besides this, the study protocol is well designed and the manuscript is well written. Only minor spelling/grammar mistakes were found (which should be addressed by the authors). Specifically, the randomization procedure is a bit unclear to me. Please clarify how this will be achieved. In terms of the power calculations the two trials aim to recruit around 3000 women, which seems sufficient based on previous findings. However, I would like to get a more in-depth explanation to the power calculations done – maybe these could be added as an appendix? Also, the choice of telemedicine equipment is not sufficiently described. Besides writing about the Microlife WatchBP Home as the choice of blood pressure device to be used, please report more in-depth on the choice of telemedicine application and the exact methodology and technologies you intend to use, e.g. how data is transferred from WatchBP device to your research database. The reference given here is “under submission”. Please clarify how you will prevent reporting error and erroneous measurement adherence during the self-measurement processes described (if using manual reading and data entry by the participants themselves). Note, that previous work has found problems with manually reported blood pressure values. Also, in terms of the qualitative interview and ethnographic studies, more information could be supplied on suggested coding and thematization methods. Besides these (minor issues) I believe this will be a very interesting and important study: I am looking forward to the results and I wish you good luck moving forward.
--	---

VERSION 1 – AUTHOR RESPONSE

Reviewer 1 Comments:

1) “The authors are to be applauded for this study protocol. I am looking forward to the results are presented.”

RESPONSE: The authors thank reviewer 1 for their comment.

Reviewer 2 Comments:

1) “Only minor spelling/grammar mistakes were found (which should be addressed by the authors).”

RESPONSE: Thank you. The authors have reviewed the manuscript and corrected any spelling or grammar mistakes.

2) “Specifically, the randomization procedure is a bit unclear to me. Please clarify how this will be achieved.”

RESPONSE: Further detail has been added to the randomisation section to clarify the procedure: “Women who agree to participate in the trial (BUMP 1 or BUMP 2) will be randomised on a 1:1 ratio either to BP self-monitoring or usual care by the recruiting researcher. An independent statistician will generate a randomisation sequence list for each trial, using permuted varying blocks and stratified by recruitment site and parity. The generated schedules were then imported to the randomisation module within the online data management system for site to carry out the randomisation. Women will be randomised by the recruiting researcher using the online data management system. Women who develop hypertension during BUMP 1 and migrate to BUMP 2 will stay in their original randomisation group. A manual telephone based back-up randomisation system will be used in the event the online system is not available.”

3) “In terms of the power calculations the two trials aim to recruit around 3000 women, which seems sufficient based on previous findings. However, I would like to get a more in-depth explanation to the

power calculations done – maybe these could be added as an appendix?”

RESPONSE: Detailed sample size calculations are already included on page 9 “sample size considerations”. These run to over 450 words and we have only made minor changes to the BUMP 1 sample size section for clarity.

“The sample size was determined using a two stage simulation process which modelled how many women would be expected to develop hypertension and how long time to detection would take in these women, using data from our pilot work in the BUMP study. Assuming 16% of women develop hypertension, and a standard deviation (SD) of 40 days to detection of raised BP in both groups, a sample size of 2262 (1131 per group) will allow detection of an effect size of 12 day’s difference in time to detection of raised BP in pregnancy between self-monitoring and control groups (the primary outcome of BUMP 1), with 90% power and 5% level of significance (2-sided) and assuming a 15% attrition rate. If the SD is 45 days, then this sample size will allow detection of a difference of 14 days with more than 90% power and if the SD is 50 days then it will be sufficient to detect a difference of 16 days in time to detection of raised BP in pregnancy also with 90% power. Of the 2262 women recruited to BUMP 1, around 362 women are expected to develop hypertension. We will recruit a minimum of 2262 women to ensure adequate power. The simulation was carried out using R 3.1.2. (<https://www.r-project.org/>).”

4) “Also, the choice of telemedicine equipment is not sufficiently described. Besides writing about the Microlife WatchBP Home as the choice of blood pressure device to be used, please report more in-depth on the choice of telemedicine application and the exact methodology and technologies you intend to use, e.g. how data is transferred from WatchBP device to your research database. The reference given here is “under submission”. “

RESPONSE: Much of the information regarding the design of the telemonitoring system and the intervention used within the BUMP Trials is covered by the paper cited which has now been accepted to the journal “Pilot and Feasibility Studies” and is in press. We have updated the reference accordingly and edited the relevant paragraph as below. We would like to avoid overlap with the information which is already in press but have still included more information in the main text of this manuscript to further describe the system itself:

“Participants can switch between the text and app system to decrease problems with connectivity e.g. poor internet connection or phone signal. Access to the telemonitoring system was designed to have secure logins for the participants, their clinicians, and the research team. This system was developed from our pilot work with appropriate theoretical underpinnings and designed by the University of Oxford, Department of Engineering Science, and appropriate functionality testing was done by the development team to test different combinations of normal and abnormal BP readings and user behaviours (e.g. poor adherence or numerous, unrequested readings) over a prolonged period followed by user testing with pregnant women.”

This accepted manuscript has been uploaded for information as it is not currently available.

5) “Please clarify how you will prevent reporting error and erroneous measurement adherence during the self-measurement processes described (if using manual reading and data entry by the participants themselves). Note, that previous work has found problems with manually reported blood pressure values. “

RESPONSE: Our previous work in pregnancy and the post-natal period suggests that women are better at transcribing blood pressure results than previously reported (e.g. SNAP-HT study found <5% error rate and only half of these affected thresholds. We will further evaluate this in our process evaluation to check that the results of the current trials are consistent with our previous work.

6) “Also, in terms of the qualitative interview and ethnographic studies, more information could be supplied on suggested coding and thematization methods.”

RESPONSE: Further information has been added to the “Process Evaluation (qualitative and quantitative)” section:

“Inductive and deductive approaches to categorising and coding the data will be combined, drawing on a priori theoretical concepts (including, for example, self-efficacy, unexpected use of technology, and cognitive participation) while remaining sensitive to themes that emerge from the data themselves. An iterative analytic process will be employed to map the range of phenomena and identify associations between themes with the aim of illuminating the mechanisms that underpin the outcomes of the intervention. QSR NVivo will be used to support the organisation and retrieval of the qualitative data.”

We thank all the reviewers for their time and hope our responses cover all their comments.

VERSION 2 – REVIEW

REVIEWER	Stefan Wagner Aarhus University, Department of Engineering, Denmark
REVIEW RETURNED	24-Dec-2019

GENERAL COMMENTS	I have reviewed the original and revised protocol (following minor revisions) for the BUMP 1&2 trials. The BUMP 1&2 trials are two NIHR funded trials seeking to assess the effectiveness and cost effectiveness of self-monitoring of blood pressure in pregnancy using a multicenter RCT approach. A very relevant research topic to pursue. Specifically, BUMP 1 assesses whether self-monitoring of blood pressure can detect raised blood pressure earlier than routine clinic monitoring which is highly relevant, while BUMP 2 assesses whether self-monitoring of blood pressure alongside usual care leads to lower blood pressure in hypertensive pregnancy. Both of these studies are highly relevant to perform as not much work has been made in this area. Also, the economical evaluation is highly relevant, as a broader home screening of blood pressure is expensive in terms of equipment and personnel resources for follow-up. Although the protocol is quite lengthy this can be explained by the combined BUMP 1&2 protocols, both of which are very ambitious with both quantitative and qualitative studies, appearing in one protocol. While this is arguably reasonable, due to the two studies being tightly connected, it also reduces the clarity of the overall protocol. There are many factors to consider. Besides this, the study protocol is well designed and the manuscript is well written. Only minor spelling/grammar mistakes were found (which was addressed by the authors in the second revision). Specifically, the randomization procedure was unclear but has been clarified. A detailing of the power calculation was requested and provided. Also, the choice of telemedicine equipment was not sufficiently described. Besides writing about the Microlife WatchBP Home as the choice of blood pressure device to be used, please I have suggested to provide more in-depth on the choice of telemedicine application and the exact methodology and technologies you intend to use, e.g. how data is transferred from WatchBP device to your research database. The original reference given was “under submission”. In the revised manuscript this has been clarified, although additional information could still be relevant to provide in an appendix, or a secondary paper reporting on the technology used. More information on how to prevent reporting error and erroneous measurement adherence during the self-measurement processes described (if using manual reading and data entry by the participants themselves) was requested, and this has been provided. Note, that previous work has found problems with
--

	manually reported blood pressure values. Also, in terms of the qualitative interview and ethnographic studies, more information was requested to be supplied on suggested coding and thematization methods, and this has been provided. Besides these (minor issues) I believe this will be two very interesting and important studies. I am looking forward to the results and I wish you good luck moving forward.
--	---